# Immunobiology and Cytokine Modulation of the Pediatric Brain Tumor Microenvironment: A Scoping Review

**DOI:** 10.3390/cancers15143655

**Published:** 2023-07-18

**Authors:** Shreya Budhiraja, Hinda Najem, Shashwat Tripathi, Nitin R. Wadhawani, Craig Horbinski, Matthew McCord, Alicia C. Lenzen, Amy B. Heimberger, Michael DeCuypere

**Affiliations:** 1Division of Pediatric Neurosurgery, Ann and Robert H. Lurie Children’s Hospital of Chicago, Chicago, IL 60611, USA; 2Department of Neurological Surgery, Feinberg School of Medicine, Northwestern University, Chicago, IL 60611, USAcraig.horbinski@northwestern.edu (C.H.); amy.heimberger@northwestern.edu (A.B.H.); 3Malnati Brain Tumor Institute of the Lurie Comprehensive Cancer Center, Feinberg School of Medicine, Northwestern University, Chicago, IL 60611, USA; matthew.mccord@nm.org; 4Division of Pathology, Ann and Robert H. Lurie Children’s Hospital of Chicago, Chicago, IL 60611, USA; 5Department of Pathology, Feinberg School of Medicine, Northwestern University, Chicago, IL 60611, USA; 6Division of Hematology, Oncology, Neuro-Oncology, and Stem Cell Transplantation, Ann and Robert H. Lurie Children’s Hospital of Chicago, Chicago, IL 60611, USA; alenzen@luriechildrens.org

**Keywords:** pediatric brain tumor, immunotherapy, microenvironment, therapeutic strategies

## Abstract

**Simple Summary:**

Pediatric brain tumors are unique from adult tumors and pose challenges due to their distinct characteristics, including differences in tumor immunology, molecular profiles, and response to various treatments. Understanding these differences is crucial for developing targeted and effective therapeutic strategies. This review delves into our current understanding of the immunobiology of various pediatric brain tumors and aims to unravel the intricate interactions between the tumor microenvironment and immune system. By identifying the essential characteristics and immunobiology of pediatric tumors, we can explore strategies to leverage the immune system for novel treatment approaches.

**Abstract:**

Utilizing a Scoping Review strategy in the domain of immune biology to identify immune therapeutic targets, knowledge gaps for implementing immune therapeutic strategies for pediatric brain tumors was assessed. The analysis demonstrated limited efforts to date to characterize and understand the immunological aspects of tumor biology with an over-reliance on observations from the adult glioma population. Foundational knowledge regarding the frequency and ubiquity of immune therapeutic targets is an area of unmet need along with the development of immune-competent pediatric tumor models to test therapeutics and especially combinatorial treatment. Opportunities arise in the evolution of pediatric tumor classification from histological to molecular with targeted immune therapeutics.

## 1. Introduction

Brain tumors are the most common solid tumors in children, with approximately 5000 new diagnosed cases per year (https://seer.cancer.gov/statfacts/html/childbrain.html accessed on 14 July 2022). Despite significant advances in surgical, radiotherapeutic, and chemotherapeutic strategies over the last several decades, brain tumors still remain the largest cause of cancer-related mortality in children [1]. Current standards of care leave patients impacted with significant long-term sequela [2]. Due to the continual need for more effective therapeutic modalities and the infiltrative nature of many pediatric brain tumors, immunotherapy represents the future horizon for adjuvant therapy.

The field of cancer immunotherapy has impacted the outcomes for many adult solid cancers. New treatment options for pediatric brain tumors can significantly lag behind those for adults [3]. Several studies have attempted to apply adult brain tumor immunotherapies in pediatric patients, but these treatments have not been particularly successful [4,5,6]. The lack of success can be attributed to differences in anatomic, developmental, genetic, epigenetic, and immunological landscapes between the adult and pediatric brain tumors. For instance, while most adult brain tumors arise from supratentorial structures, 60% to 70% of pediatric brain tumors, such as pilocytic astrocytoma (PA), medulloblastoma, and ependymoma, develop in the infratentorial compartment. Furthermore, pediatric brain tumors tend to be genetically very different from adult brain tumors, featuring alterations, like histone mutations in high-grade gliomas (HGGs) or BRAF molecular alterations in PA. In general, these unique genetic differences require diverse approaches to treatment, relative to those of adult brain tumors. Although several studies describe the impact of host immunity on the survival of pediatric patients, there are clear gaps in the literature defining the immunobiology of pediatric brain tumors [4,5,6]. The goals of this scoping review are as follows: to identify the gaps in the characterization of the immunobiology with a specific focus on therapeutic targets, potentially prioritize available therapeutics, and identify opportunities for treatment with the available compendium.

## 2. Methods

### Search Strategy

A scoping review was conducted according to the PRISMA-SR (Preferred Reporting Items for Systematic Reviews and Meta-Analysis Extension for Scoping Reviews) guidelines. PubMed MEDLINE, Embase, and Scopus databases were searched on 14 July 2022, using keywords, such as “brain tumor”, “pediatric”, and “immunology”. No restrictions on the date, study type, or language were applied. Full search terms are displayed in Appendix A. The protocol was not registered. Upon review, duplicates were eliminated via automatic deduplication in Endnote X9 (Clarivate Analytics, London, UK). All remaining articles were screened by title and abstract for relevance. Articles progressing to full-text review were screened for final inclusion based on prespecified inclusion and exclusion criteria. Inclusion criteria were written in or translated into the English language, with full-text available, studying any type of pediatric brain tumor and reporting any clinical outcomes or immunological properties of pediatric brain tumors. Exclusion criteria included conference abstracts, case reports, narrative reviews, systematic reviews, and meta-analyses. A second reviewer replicated the search strategy, and disagreements were reconciled via consultation with a third reviewer.

## 3. Results

Using PubMed MEDLINE, Embase, and Scopus databases, 52 articles were identified in the literature related to pediatric brain tumor immune biology (Figure 1A; Appendix A). Relevant information was gathered from selected studies, including the study design, bibliographic data, pediatric brain tumor type(s), immune-related interventions (such as immunotherapy), study outcomes, and clinical outcomes. Any discrepancies were resolved through discussion or by an additional reviewer if necessary. The extracted data were synthesized and analyzed to provide an overview of the current state of knowledge on the intersection of pediatric brain tumors and immunology, including potential immunological targets and strategies for improving patient outcomes.

## 4. Discussion

### 4.1. General Themes of Pediatric Brain Tumor Immunobiology

The pediatric brain tumor microenvironment consists of an interconnected population of neurons, astrocytes, microglia, and oligodendrocytes residing within a unique extracellular matrix (ECM) [7]. Given this complexity, studies are typically focused on the following themes: (1) the general state of immunosuppression in pediatric patients (48%); (2) immune surveillance in the brain (15%); (3) the mutational burden of pediatric brain tumors (13%); and (4) differences between adult and pediatric glioma immune profiles (10%) (Figure 1B).

### 4.2. Differential Immune Surveillance within Pediatric Brain Tumors

For diffuse midline gliomas, CD3 infiltration frequency is similar in both adults and pediatric patients, whereas CD8 expression may be greater in adults [8]. Some investigators have noted that pediatric brain tumors exhibit a less immunosuppressive tumor microenvironment when compared to adult brain tumors [9]. When investigating molecular subtypes within pediatric HGG, investigators found that immune-suppression markers were predictive of survival in only certain molecular subgroups. In K27-mutated tumors, PD-L1 and CTLA-4 confer a worse prognosis, while this effect was not detected in patients with G34-mutated gliomas. The G34-mutated tumors have been found to rely on the TGFB1 and HAVCR2 (TIM3) pathways for immune evasion [10] indicating that the underlying genetic makeup of the glioma may influence preferential dominant immune-suppressive mechanisms.

One study has shown that pediatric brain tumors can be classified into five different proteomic immune signatures. The first group consists of the predominating infiltration of macrophages, microglia, and dendritic cells, epithelial–mesenchymal transition (EMT), and the presence of adenosine-mediated immune suppression. The second group is characterized by upregulation of the immune-suppressive glutamate signaling pathway. These groups include a mixture of low-grade gliomas (LGGs) and HGGs. The third group, consisting mostly of craniopharyngiomas, was characterized by an increase in EMT, CTLA-4, and PD-1 expression. The final two groups consist of low levels of immune infiltration and exhibited the upregulation of WNT signaling [11].

In an analysis spanning across brain tumor types, such as PA, ependymoma (EPN), glioblastoma (GBM), and medulloblastoma (MB), PA and EPN had a higher frequency of infiltrating immune cells, including activated myeloid cells, compared to GBM, MB, or normal tissue [12]. One group has reported that LGGs had higher T cell density when compared to HGGs. However, within LGG, T cell infiltration was dependent on the cancer lineage with pleomorphic xanthoastrocytoma (PXA) and ganglioglioma containing relatively higher T cell densities [13]. Another study found that PXA had significantly higher CD8+ T cell infiltration than gangliogliomas [14]. A comparison between GBM and MB showed that the latter had a particularly low amount of tumor immune cell infiltration [15]. These data indicate that each type of brain tumor is associated with varying tumor-infiltrating lymphocyte levels [16,17]. In a cohort of MB, PNET, and astrocytomas, 76% of these tumors were positive for CD8+ T cells, 85% contained CD4+ T cells, and 97% contained macrophages, but these constituted only 1–10% of the total cells [18]. DIPG patients have decreased NK cells and increased B cells in the peripheral blood when compared to control blood samples, suggesting that these two immune cell populations may be differentially trafficked during DIPG growth [19]. Notably, even within specific glioma pathologies, such as PA, in which there is robust immune infiltration, this can be highly variable [20].

There are likely multiple factors contributing to the degree and types of immune infiltration throughout the tumor microenvironment (TME) that include but are not exclusive to the following: (1) the presence of an immunogenic antigen; (2) immune chemokine expression; (3) the disruption of the blood–brain barrier; (4) genetic and epigenetic features; and the (5) types of immune suppression that predominant within a given malignancy. Notably, the presence of immune cells within the TME does not imply functionality because they can be either anergic or exhausted. As such, future studies should include functional assessments. Although these studies effectively convey the challenge of heterogeneity within these types and subtypes of tumors, there is still a marked need for further granularity on the functions and states. It is still unclear how various immune cells are distributed across the tumor and TME and whether their roles in promoting or suppressing the tumor are conserved across different regions. Given this, novel approaches are required to gather data beyond simply the quantification of various immune cells in each tumor. Instead, targeted investigations of the immune cell distribution and signatures across various locations within the TME are needed to fully understand the role of these cells and how they can be best manipulated to generate an effective anti-tumor immune response.

### 4.3. The Role of Immune Suppression in Pediatric Brain Tumors

It has been postulated that pediatric brain tumors may actually be less immunosuppressive than adult brain tumors, suggesting that some adult therapies that may not have been successful should not be discarded as viable treatment options in children and adolescents [9]. A key distinguishing factor between adult and pediatric immunobiology is the generally immature state of pediatric immune systems. The transition from fetal to postnatal life requires an associated transition from immunosuppression to immunological responsiveness [21]. Given the evolving state of the pediatric immune system, the effects of these factors on brain tumor immune profiles between children and adults also await further in-depth studies. Studies conducted thus far have detailed mechanisms that are known to play a role in neonatal immunity within the brain, such as increased Th2 CD4+ cells, reduced CD8+ T cells and interferon (IFN)-gamma responses, immaturity of dendritic cells (DC), and increased IL-10 secretion by antigen-presenting cells. None, however, have described the potential effects of these cells on brain tumor growth [22,23,24,25]. Immunosuppressive mechanisms, such as adenosine and myeloid-derived suppressor cells (MDSCs), increase immediately after birth, contributing to the tolerogenic immune status of pediatric patients. However, these mechanisms have mostly been studied in adult brain tumors [26,27,28,29]. One study investigated the systemic immune profiles of children with brain tumors to understand if there was a common pattern. Only MB patients had a unique serum cytokine profile characterized by high VEGFA and IL-7 and low IL-17A and TNF-β [30].

Unique to the CNS are microglia, which are macrophage-like cells that reside in the brain parenchyma [31]. Throughout brain development, microglia regulate a number of immune chemokines, such as CXCL12 and CXCR4 [32], which could create a specific microenvironment for tumor cells to emerge [33]. Although studies have outlined the plasticity of microglia in certain diseases, the identification of specific pro-tumor subtypes of microglia, as well as mechanisms by which these subtypes arise to promote brain cancer, has not been studied extensively in children [34].

### 4.4. Pediatric Brain Tumors Typically Have a Very Low Mutational Burden

Most pediatric brain tumors have a low tumor mutational burden (TMB) [35]. Despite this, there tends to be a higher occurrence of epigenetic changes compared to adult brain tumors. This distinction can be attributed to the following reasons: (1) certain tumors, like MB, originate during embryonic development; (2) developmental pathways in children may experience dysregulation, contributing to tumor growth; and (3) children have a shorter duration of exposure to environmental factors that act as carcinogens [36]. In the comprehensive genomic profiling of 723 pediatric brain tumors, low TMB was present in 92% of cases [37]. This observation also holds for other rare cancers, such as malignant rhabdoid tumors [38]. Based on grade, both LGG and HGG pediatric brain tumors have a low TMB; however, 6% of HGGs are hypermutated with greater than 20 mutations/Mb [39].

This low TMB presents a unique challenge for immunotherapeutic treatment. Some immunotherapies rely on targeting a specific tumor antigen. Since these develop from mutations, the lower TMB means that these antigenic targets are less prevalent in pediatric brain tumors and may confer a decreased chance of a response to some immune therapies [40]. This has been further corroborated by an analysis of tumor antigen precursor protein profiles, in which adult gliomas expressed 94%, whereas only 55–74% of pediatric gliomas did so [41]. Some pediatric tumors, such as MB, have more antigens and NK infiltration, which positively correlate with prognosis, suggesting that these tumors may be more easily targeted with immune therapeutics [42,43,44]. In adult gliomas, a low TMB may not be a biomarker of a response to some immunotherapies, such as virotherapy [45]. Although virotherapy has been evaluated in pediatric glioma trials, it is unclear if there is a therapeutic benefit [46,47].

### 4.5. Immunoediting as a Framework for Pediatric Brain Tumor Immunology

The concept of “tumor immunoediting” has been widely used as a framework for understanding how tumors still develop despite the myriad of immune responses that they elicit (Figure 2). While our understanding of pediatric brain tumors remains limited, following this framework allows for a systematic approach in investigating the complex immunological interactions occurring within the developing pediatric brain that may lead to potential immunogenicity and the immune-evasion mechanisms of these tumors. Although significant challenges exist in comprehending the intricate dynamics of pediatric brain tumor immunology, the adoption of the immunoediting concept offers a robust foundation advancing innovative therapeutic strategies in each different phase of this framework. The general cancer immunoediting concept has three major phases by which innate and adaptive immune cells respond to tumor cells: (1) elimination; (2) equilibrium; and (3) escape [48,49,50]. In the elimination phase, tumor cells that escape the non-immune mechanisms of tumor suppression are recognized and eliminated by innate immune cells. The incomplete eradication of these tumor cells drives the next phase, equilibrium. During this phase, tumor cells enter a state of dormancy, where they can evolve to become immunosuppressive or modulate tumor-specific antigens to escape immune recognition. Through this, there is a balance between the recognition of tumor cells by the adaptive immune system and tumor mutations. Finally, during the third phase, escape, the multitude of tumor mutations caused by immune pressure selects for an immune-resistant tumor phenotype. The tumor is then able to generate an immunosuppressive environment by escaping recognition by anti-tumor immune cells and generating pro-tumor immune responses [48,49,50].

The first step of the elimination phase relies heavily on the innate immune system recognizing tumor cells through the recognition of damage-associated molecular patterns (DAMPs), rather than tumor-specific antigens [50,51]. DAMPs in brain tumors include uric acid, heat-shock proteins, ligand transfer molecules induced by CpG DNA, and extracellular matrix derivatives that serve as ligands for toll-like receptors (TLR). The recognition of these DAMPs causes the activation of pro-inflammatory responses, the maturation of dendritic cells, T cell antigen presentation, and the release of pro-inflammatory signals, which lead to the recruitment of immune cells that recognize tumor cells and release IFN-γ [4,52]. TLR7 expression has been found to be a prognostic factor in MB patient survival [53]. The predominate cells participating in this activity are thought to likely be the CNS-resident microglia. There is selective enrichment of microglia/macrophage-related genes in pediatric HGG of the mesenchymal subtype [54]. Although these studies reveal the importance of a heightened TLR response in immune cells to recognize and eliminate tumor cells, TLRs are also expressed on brain tumor cells, thereby playing dual roles in eliciting anti-tumoral and pro-tumoral responses [55]. Some clinical trials have suggested signals of a clinical response using TLR agonists [56]. However, others have shown that the inhibition of pro-tumor TLR signaling may suppress glioma growth [57,58]. Although there has been a full characterization of the different TLRs and their pro-tumor or anti-tumor roles among various adult brain tumors, TLR expression in pediatric brain tumor types is still being studied [56]. Despite the limited number of studies on TLR immunotherapy against brain tumors, the use of TLR agonists in conjunction with immune checkpoint inhibitors has been shown to effectively induce cytotoxic T cell responses to suppress cancers [59].

The release of IFN-γ from these innate immune cells results in limited killing of the tumor cells through various anti-proliferative, anti-angiogenic, and apoptotic effects. As immature dendritic cells start to mature, antigen-loaded dendritic cells migrate to the cervical lymph nodes, where they present tumor antigens to CD4+ and CD8+ T cells. There, presentation to naive T cells allows for the differentiation, maturation, and clonal expansion [4,52]. Thereafter, the immune effector cells migrate and infiltrate the tumor. An antigen-presenting event within the tumor is likely important for immunological recognition of the cancer [16,17]. During the equilibrium stage, there is antigen modulation ultimately resulting in antigen loss, which has been described in several clinical trials [60,61] and low levels of immunogenicity mediated by the lack of MHC expression and co-stimulatory molecules that trigger immunological anergy.

During immune escape, immunogenic tumor cells are no longer detected by the immune system because of the following: (1) the loss of immunogenic antigens [62,63]; and (2) downregulation of the expression of MHC [64,65,66], or the process of antigen presentation is downregulated through the cGAS/STING pathway [67]. Specifically, in pediatric brain tumors, the downregulation of MHC-I and CD1d has been documented [68,69]. During the escape phase, the TME of pediatric brain tumor patients is notable for the marked immunosuppressive cytokines. In MB, a mutated mTOR pathway leads to increased IDO1 expression [70], which triggers the expansion of regulatory T cells enabling tumor cells to grow [71]. IDO inhibitors have been tested in a wide variety of oncological indications, including adult glioblastoma, but the Phase II results have not yet been released [72]. Glutamate has been noted to be aberrantly expressed in multiple cancers [73] and can promote immune-evasion mechanisms via immunosuppressive cytokine production in certain types of pediatric brain tumors [74]. Adenosine is another immune-suppressive pathway that can be operational in these tumors [75,76]. TGF-β has been extensively documented as being immunosuppressive in the TME of adult gliomas, but also in MB [77,78]. When TGF-β signaling is blocked, regulatory T cells are decreased and there is an increased capacity of CD8+ T cells to carry out cytotoxic functions [77]. TGF-β also functions in MB to antagonize NK anti-tumor functions [79], which can be therapeutically manipulated for anti-glioblastoma activity [80]. Tumor-derived exosomes (TEXs) can also mediate immune suppression in pediatric brain tumors [81]. MB-secreted TEXs have been shown to inhibit IFN secretion from T cells in a dose-dependent manner [82]. However, these TEXs can also be immune-stimulatory [83], so the key determinant of immune modulation is likely dependent on the exosomal content.

Monocytes can become polarized to support tumor growth and suppress the anti-tumor immune response. Pro-tumor-polarized tumor-associated macrophages (TAMs) in pediatric brain tumors have been identified in a wide-variety of pediatric brain tumors [84]. This polarization may occur through the nuclear factor-kappa B (NF-κB) pathway, which was found to be increased in posterior fossa group A (PFA) ependymomas [85]. The NF-κB complex promotes the transcription of many cytokines, including IL-6, which, when chronically secreted, maintains the immunosuppressive environment thorough the polarization of infiltrating monocytes [85,86]. Platelet-derived growth factor subunit B (PDGFB) has also been implicated as playing a role in immunosuppressive macrophages in pediatric HGG, in which shorter murine median survival was accompanied by an increase in the TAM infiltration of PDGFB-driven tumors [87]. The mechanism of TAM recruitment and differentiation in pediatric craniopharyngiomas may be attributed to both the IL-8 and IL-6 cytokines [88]. IL-8 promotes myeloid-derived suppressor cell recruitment to the tumor, while IL-6 activates the JAK/STAT pathway—a key hub of tumor-mediated immune suppression [89]. STAT3 inhibitors are in clinical trials for pediatric patients with brain tumors (NCT04334863). The expression of immune-suppressive macrophages within MB is markedly heterogeneous, but there could be enrichment within the SHH-group [90]. Research into the functions of TAMs across subtypes of MB will be needed before the implementation of therapeutics targeting this population since the TAM may also have anti-tumor properties [91]. Nonetheless, in clinical trials for adult brain tumors, targeting TAM activity, recruitment, and/or polarization may be beneficial especially when used in combination with immune checkpoint inhibitors [92].

Under normal physiological conditions, the activity of T cells is controlled, in part, through interactions between programmed cell death-1 (PD-1) and programmed death-ligand 1 (PD-L1) [93]. Targeting the PD-1/PD-L1 axis has revolutionized the field of cancer immunotherapy [94], especially for cancers that have a high TMB and immune infiltration [95]. Although the expression of this immune-suppressive axis is common in many solid cancers, the expression in pediatric tumors is less common [96,97] and can be markedly heterogeneous [98]. In supratentorial extra-ventricular ependymomas, PD-L1 expression was a negative prognosticator for progression-free survival [99]. Ependymoma that harbored a RELA fusion (ST-RELA) had high PD-L1 expression on both the tumor cells and myeloid cells. T cell exhaustion was confirmed through PD-1 detection on both CD4+ and CD8+ T cells, as well as their inability to secrete IFNγ upon stimulation [100]. Other profiling initiatives in pediatric CNS tumors [101,102] indicate that the infrequent expression of the PD-1/PD-L1 axis is not the sole mechanism for the generation of T cell exhaustion and that other mechanisms of immune suppression are operational. As such, the use of this type of immunotherapy strategy likely needs to be considered in select subsets of these patients.

Tumor cells avoid immune-mediated apoptosis through the upregulation of anti-apoptotic factors, downregulation of death receptors, or granule exocytosis. The best characterized death receptors are CD95 (APO-1/Fas), TNF receptor 1 (TNFRI), TRAIL-R1, and TRAIL-R2 [103,104,105,106]. Alternatively, tumor cells upregulate the dominant-negative Fas-associated death domain (FADD) or intracellular FADD-like inhibitory protein (FLIP) [103,104,105,106]. The majority of childhood glial tumors, particularly astrocytomas, express the death receptor CD95 [18]. The granzyme intracellular pathway is the other avenue for apoptosis. In this mechanism, proteases are released into the immune synapse, where they travel through the perforin pores into target cells generating pro-apoptotic effectors, granzyme A and granzyme B. Tumor cells are able to prevent elimination from the perforin/granzyme pathway by abnormally eradicating granzyme expression or increasing intrinsic granzyme inhibitors, called serpins [107]. In MB and in primitive neuroectodermal tumors (PNET), the granzyme inhibitors SERPINB1 and SERPINB4 were acquired in 23% and 50% of the MB, respectively, while PNETs expressed SERPINB9, SERPINB1, and SERPINB4 in 29%, 29%, and 57% of the tumors [68,69].

### 4.6. Evolution of Pediatric Brain Tumor Classification

In recent years, the pathologic classification of both adult and pediatric brain tumors has increasingly used genetic, cytogenetic, and epigenetic (i.e., DNA methylation profiling) features. Histology still provides useful information but has limitations. Two tumors with a similar morphology may have completely different biology and behavior and require completely different therapies. The 2021 WHO CNS tumor classification system is based on integrated molecular diagnostics, combining relevant morphologic and molecular features to identify tumors and determine key prognostic and predictive information. Distinct molecular groups of MB have been recognized since at least 2012 [108], and an integrated molecular diagnostic approach was recommended in the 2016 WHO CNS tumor classification system [109]. The 2021 system takes a similar approach for pediatric-type gliomas. For example, the mutation in histone H3 at G34 is a well-known molecular driver in pediatric-type high-grade hemispheric gliomas [110,111]. This mutation is now a formal diagnostic criterion for the tumor class [112]. The WHO has also defined several new classes of pediatric-type gliomas, based on molecular drivers. These include diffuse astrocytoma, MYB- or MYBL1-altered; diffuse low-grade glioma, MAPK pathway-altered; polymorphous low-grade neuroepithelial tumor of the young (BRAF mutations and FGFR2 fusions); and infant-type hemispheric glioma (alterations in NTRK, ROS1, ALK, or MET) [112,113]. Furthermore, DNA methylation profiling [114,115] has been incorporated into the diagnostic criteria for many types of adult and pediatric brain tumors. In some instances, DNA methylation data can make distinctions that are otherwise difficult to ascertain, such as between PF groups A and B in ependymoma [112,116]. Because of the evolution of the aforementioned pathology changes, the analysis of immune biology as a function of molecular drivers is eminently emerging. These changes in characterization will likely drive patient selection for future clinical trials, including Basket Trials. As opposed to enrolling patients with a given cancer-cell-lineage diagnosis, molecular driver classification will likely inform selection and/or stratification.

### 4.7. The Paucity of Pre-Clinical Pediatric Brain Tumor Models

Currently utilized pediatric brain tumor models include genetically engineered murine models (GEMMs) and patient-derived xenografts (PDX) that mostly consist of HGG and medulloblastomas. Several biobanks exist with established PDX models that are available upon request. Tumor cell lines available through these resources have accompanying histopathology, whole-genome sequencing, RNA-sequencing, and DNA array sets providing researchers with a characterization for the selection of cell lines. While these biobanks have expanded the available pediatric brain tumor models, there are still limitations for other cancer lineages, including clinical and molecular annotation within these repositories. The major limitation with PDX models is the immune-incompetent background that hampers the ability to investigate the immune composition and interaction with the tumor and to screen and test immunotherapies. Recent developments of humanized mice that are reconstituted with human bone marrow provide a new avenue to test immune therapies, but these models are associated with a high cost, preventing large-scale drug testing, and it is still unknown how well these mice recapitulate the human biology. GEMMs can overcome some of these inherent limitations of PDX models since they have a native immune system and intact microenvironments. GEMMs enable researchers to study the early stages of tumor initiation, which is particularly relevant for pediatric brain tumors, since the source of embryonal tumors is still unknown and may arise from embryonic or fetal cells remaining in the CNS. GEMMs are created by mutating key pathways, such as EGFR, PDGF, NF1, and TRP53, which are altered in human gliomas [117,118]. These models have inherent limitations, including the maintenance of breeder colonies. Additionally, gliomas have alterations in several pathways, and choosing the correct GEMM can be difficult. Finally, these models may have the correct mutations and genetic alterations; however, several overlapping yet distinct tumor types may share similar alterations. For example, MAPK-driven pediatrics encompass several cancer lineages with unique characteristics that will likely require different treatment modalities.

## 5. Conclusions

### 5.1. Overview of Findings, Future Perspectives, and Implementations

There are a number of challenges in the development of immunotherapy for pediatric brain tumors (Figure 3). The recent successes in adult tumor immunotherapy for solid cancers has revealed how the understanding of immunobiology may lead to effective treatment options. In order to achieve this in pediatric patients, it will first be necessary to acquire a better understanding of the pediatric brain tumor immune landscape.

This review synthesizes what is currently known about various pediatric brain tumors in an attempt to shed light onto the potential efficacy of certain treatment modalities. For diffuse midline gliomas, studies have shown that these tumors exhibit similarities in the CD3 infiltration frequency between adults and pediatric patients but that the expression of CD8 is greater in adults, suggesting potential differences in the immune response. Pediatric brain tumors, including DMG, have also been observed to have a less immunosuppressive tumor microenvironment compared to adult brain tumors, providing a unique opportunity for targeted immunotherapies. Thus, developing strategies that enhance CD8+ T cell responses and overcome immune-suppression mechanisms involved in DMG—such as TGF-β and IL-10—could significantly improve treatment outcomes.

As for LGGs, they have been found to exhibit a higher T cell density compared to high-grade gliomas. The differences in T cell infiltration between LGG subtypes, such as PXA and ganglioglioma, suggest the need for subtype-specific approaches. Targeting specific immune cell populations and enhancing T cell responses tailored to individual LGG subtypes could be explored for more effective treatments. Furthermore, other immune evasion mechanisms in LGG include the upregulation of immune checkpoint molecules, such as PD-L1 and CTLA-4, within the tumor microenvironment. Thus, combining immune checkpoint inhibitors with strategies that enhance T cell activation, such as immune-stimulatory cytokines or adoptive T cell therapy, may improve anti-tumor immune responses in LGG.

GBM has been shown to exhibit low tumor immune cell infiltration compared to other tumor types, such as MB. The higher presence of antigens and natural killer (NK) cell infiltration in MB, along with their positive correlation with prognosis, suggests the potential of NK cell-based therapies. Developing strategies that boost NK cell functions and antigen presentation in GBM could enhance immune responses against these tumors. Additionally, other immune evasion mechanisms in GBM and MB include the upregulation of immunosuppressive molecules, such as IDO. Targeting these molecules with specific inhibitors or combining immune checkpoint inhibitors with NK cell-based therapies may improve the efficacy of immunotherapies in GBM and MB.

For ependymomas, the higher frequency of infiltrating immune cells, including activated myeloid cells, in EPN compared to other tumor types indicates the possibility of targeting these immune cell populations. However, the further characterization of immune cell populations in EPN is necessary to understand their functional significance and potential immune-evasion mechanisms. In addition to myeloid cells, the presence of TAMs and Tregs in the EPN microenvironment suggests potential immunosuppressive roles. Strategies that modulate these immune cell populations, such as TAM-targeting therapies or Treg depletion, could enhance anti-tumor immune responses in EPN.

The unique immune signature observed in craniopharyngiomas, characterized by an increase in epithelial–mesenchymal transition, CTLA-4, and PD-1 expression, suggests the potential for targeting these immune checkpoints. Along with this, the activation of the Wnt/β-catenin pathway, commonly found in craniopharyngiomas, may also majorly contribute to immune evasion, implicating the importance of this pathway as a target in combination with immune checkpoint inhibitors.

Ultimately, despite the abundance of articles on tumor immunology and immunobiology, very little has been studied in the domain of pediatric brain tumors. Understanding the immunobiology of pediatric brain tumors, including the lower tumor mutational burden and the immunoediting process, is crucial for developing these effective therapeutic strategies. Further research into the mechanisms driving immune suppression, immune escape, and immune resistance within these tumor types will be essential for discovering novel drug targets and designing combination therapies that can overcome these challenges. Additionally, exploring the role of epigenetic changes in tumor growth may uncover new avenues for targeted therapies in pediatric brain tumors.

More specifically, some reports have quantified various immune lineages but typically have not delved into the distribution across the TME, especially as a function of regions of BBB permeability and tumor genetic heterogeneity. Given the variety of pro-tumor and anti-tumor roles of the various immune cell populations, the general quantification of immune cells is insufficient. Rather, more systematic and extensive profiling is needed before prioritizing the available therapeutic compendium. Efforts directed at single-cell RNA sequencing of the immune compartment along with spatially resolved transcriptomic sequencing would more effectively characterize the heterogeneous distribution of various immune cells across the TME, while also capturing relevant functional signatures. Another major hurdle is the lack of pre-clinical models in pediatric brain tumors, making potential immunotherapies difficult to evaluate. Given this, greater efforts are also needed in developing immune-competent pre-clinical models suitable for translational studies to address the unmet need for adequate therapeutic options in pediatric brain tumor patients.

### 5.2. Key Areas for Future Investigation

Determine the most frequent immune-modulatory targets that are specific to pediatric gliomas to prioritize the available therapeutic compendium.

Consider treatment strategies based on molecular alignment as opposed to the histological diagnosis.

Develop low-grade glioma models in which therapeutic modalities can be rigorously tested.

## Figures and Tables

**Figure 1 cancers-15-03655-f001:**
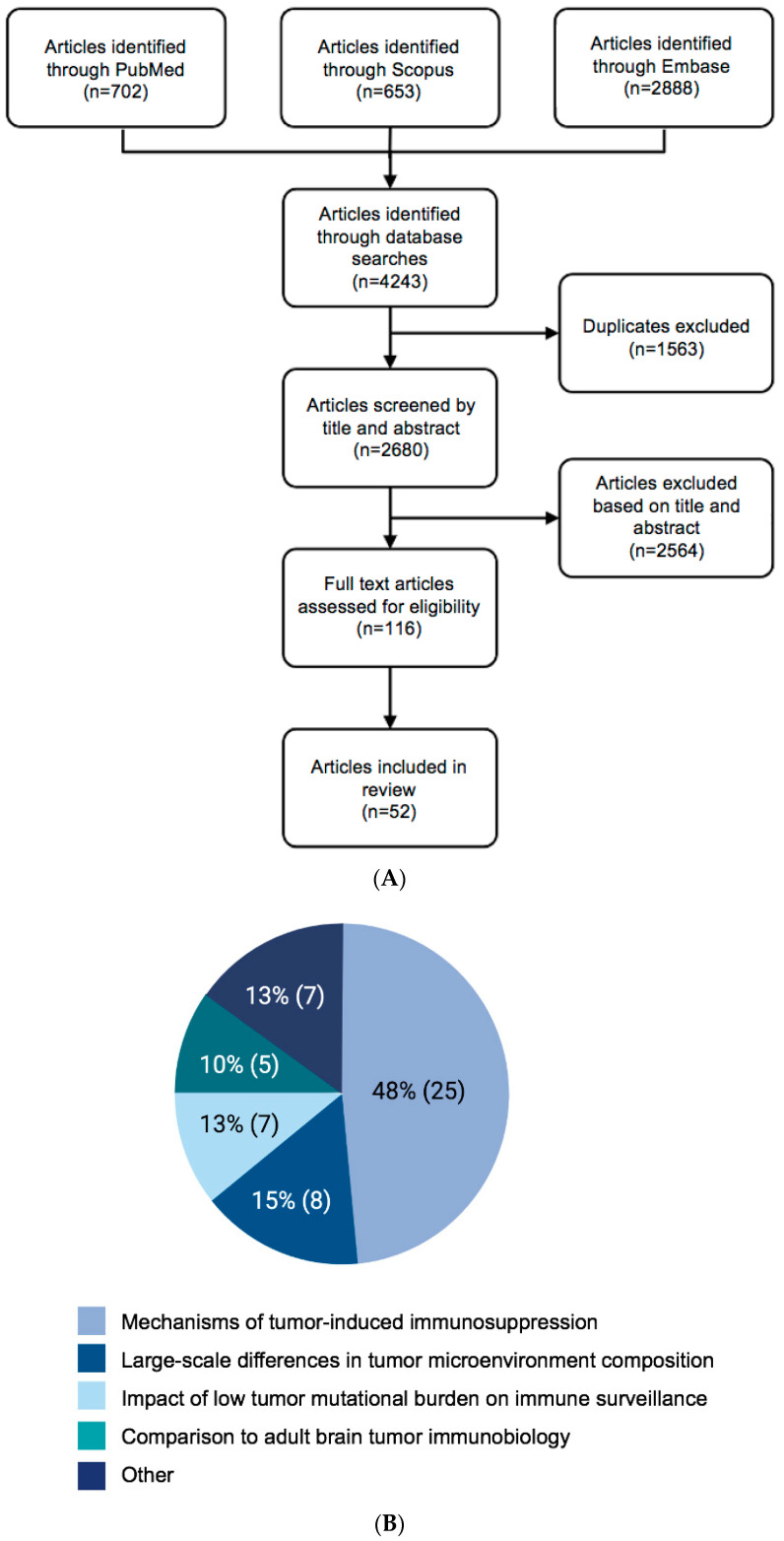
(**A**) PRISMA flow chart and (**B**) literature characteristics.

**Figure 2 cancers-15-03655-f002:**
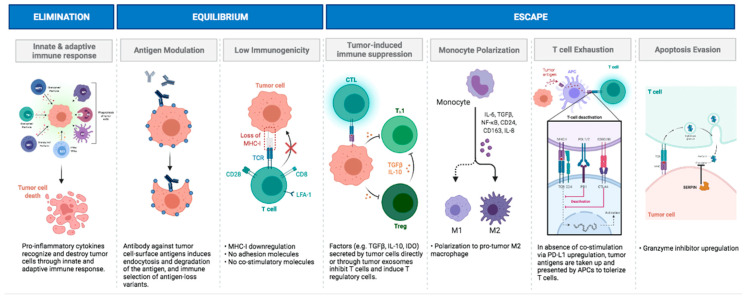
Immunoediting as a framework for pediatric brain tumor immunobiology.

**Figure 3 cancers-15-03655-f003:**
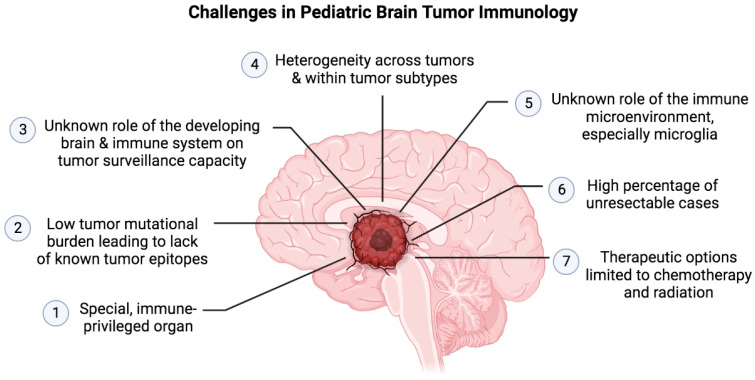
Challenges in Pediatric Brain Tumor Immunology.

## Data Availability

Data sharing not applicable.

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
