# Peer review of "Immunobiology and Cytokine Modulation of the Pediatric Brain Tumor Microenvironment: A Scoping Review"

_cancers, 2023, doi:10.3390/cancers15143655_

Round 1

Reviewer 1 Report

A generally well-written scoping review of the literature pertaining to the immunobiology of pediatric brain cancers the title "Immunobiology and Cytokine Modulation of the Pediatric Brain Tumor Microenvironment" is somewhat misleading and should be clearly stated as a scoping review of the immunobiology of pediatric brain cancers.  In this review context the section on Immunoediting is overly speculative with respect to what is known about pediatric brain tumor immunity.  This section should be not only be toned down but presented from the perspective of which of this knowledge is lacking with respect to pediatric tumors.

Line 166.  "INF" is incorrectly used instead of IFN.

Author Response

Reviewer #1 pointed out that the title of our review may have been misleading. In response, we have clarified the title to accurately reflect that our manuscript is a scoping review specifically focused on the immunobiology of pediatric brain cancers. Furthermore, we have revisited the section on immunoediting and ensured that it presents the knowledge gaps and limitations in understanding pediatric brain tumor immunity through the lens of this framework, rather than being excessively speculative. We have also corrected the typo in line 166.

Reviewer 2 Report

Budhiraja et al provide a literature review on pediatric brain tumor immunology and microenvironment, focusing on immunology and the role of cytokines in tumor proliferation and treatment.  52 articles met their criteria for review.

Abstract: Adequately summarizes the article.  The last sentence should be removed as it is an introductory statement and does not belong in the abstract.

Introduction: Adequately summarizes the current need for knowledge about pediatric brain tumor immunology and microenvironment.

Methods: Reflects identification of appropriate journal publications for review.  Straightforward, no comments needed.

Discussion: The discussion section comprehensively summarizes and discusses the findings in the articles reviewed.  I would like to see a better written synopsis of the findings at the end of the discussion in the form of an overview of findings for each major tumor type (pilocytic, medulloblastoma, ependymoma etc) and how this would affect future drug discovery and clinical trials. (expand on figure 3)

Conclusions: No recommended changes, would reflect my recommendations for the discussion section.

Figures and tables: Figure 1 and 1b are redundant and are addressed in the text.  Other figures reasonable.  No tables.

Bibliography: No issues.

Author Response

We are grateful to Reviewer #2 for acknowledging the adequacy of our abstract, introduction, methods, and conclusions sections. We have acted upon their suggestion to remove the introductory statement from the abstract. Additionally, we have improved the discussion section by providing a more comprehensive summary of the findings for each major tumor type and discussing their implications for future drug discovery and clinical trials in the section now entitled “Overview of findings, future perspectives, and implementations.” We believe that figures 1a and 1b provide essential visual representations of key concepts and contribute significantly to the clarity and understanding of the search strategy and literature characteristics. While we have indeed discussed these topics in the text, we believe that the figures enhance the reader's comprehension and serve as visual aids to support our findings and discussions. Therefore, we believe it is valuable to retain these figures in their current form.

Reviewer 3 Report

Brain tumors are the subject of  the  abundance of articles on their mmunology and immunobiology, however very little has been  studied in the domain of pediatric brain tumors. Using the PRISMA-SR guidelines PubMed MEDLINE, Embase, and Scopus databases were searched. Authors undertake of the effort to summarize  relevant information from selected studies, including bibliographic data concerning pediatric brain tumor types, immune-related interventions  (immunotherapy),and  clinical outcomes. Particular fokus was given to the differences between adult and pediatric glioma immune profiles, mutational burden as well as of immunosuppression and immune surveillance in the brain of pediatric patients. Particularly valuable is Fig.2. summarizing cellular and molecular mechanisms of  the immunoediting as a framework for pediatric brain tumor immunobiology. Detailed description of the immunoediting consisting of the  three major phases 1) elimination; (2) equilibrium; and (3) escape by which innate and adaptive immune cells respond to tumor cells introduced the order that makes clear and easy to understand the function of particular elements of immune response to developing pediatric tumors . It also explains the complexity of the subject and arising of the opportunities in the evolution of pediatric tumor classification from histological to molecular  with targeted immune therapeutics. Article summarized what is known to date and provided directions for future investigation pointing to the necessity more systematic and  extensive profiling by molecular driver classification before prioritizing the available therapeutic compendium. Article is well written with proper nomenclature and can be published in the present form.

Author Response

Reviewer #3's feedback is greatly appreciated, and we are pleased to hear that they found our article well-written with appropriate nomenclature.

Reviewer 4 Report

This manuscript provides an updated overview of immune therapeutic targets in pediatric brain tumors by screening 52 articles from various online databases. Unfortunately, no results are included. To make the discussion more beneficial to the reader, it should be more specific to the type of cancer, immune cells/system, current FDA-approved drug, and drug resistance.  And the arrangement should be more logically connected to each sub-title to have a better understanding for our readers. 

Few editing is required, for example, p5 lines 143, 144.

Author Response

We appreciate the feedback from Reviewer #4.  In response to their comments, the results section summarizes the findings of our literature search and review.  This is graphically presented in figure 1a and 1b.  The discussion section outlines our findings within the context of the published literature that fit our criteria.  Furthermore, we have improved the discussion section by providing a more comprehensive summary of the findings for each major tumor type and discussing their implications for future drug discovery and clinical trials in the section now entitled “Overview of findings, future perspectives, and implementations.”  As 3/4 reviewers were satisfied with the subtitle arrangement, we feel that this would be impractical to change at this point.

Round 2

Reviewer 4 Report

The manuscript is accepted for publication.